# Estimating the Surface Fuel Load of the Plant Physiognomy of the Cerrado Grassland Using Landsat 8 OLI Products

**Micael Moreira Santos** [1,*], **Antonio Carlos Batista** [1], **Eduardo Henrique Rezende** [2], **Allan Deyvid Pereira Da Silva** [2], **Jader Nunes Cachoeira** [2], **Gil Rodrigues Dos Santos** [3], **Daniela Biondi** [1] and **Marcos Giongo** [2]

1   Department of Forest Sciences, Federal University of Paraná, Curitiba 80210-170, Brazil
2   Environmental Monitoring and Fire Management Center—CeMAF, Federal University of Tocantins, Gurupi 77410-530, Brazil
3   Department of Phytopathology, Federal University of Tocantins, Gurupi 77410-530, Brazil
*   Correspondence: micaelmoreira@ufpr.br

**Abstract:** Techniques and tools meant to aid fire management activities in the Cerrado, such as accurately determining the fuel load and composition spatially and temporally, are pretty scarce. The need to obtain fuel information for more efficient management in a considerably heterogeneous, biodiverse, and fire-dependent environment requires a constant search for improved remote sensing techniques for determining fuel characteristics. This study presents the following objectives: (1) to assess the use of data from Landsat 8 OLI images to estimate the fine surface fuel load of the Cerrado during the dry season by adjusting multiple linear regression equations, (2) to estimate the fuel load through random forest and k-nearest neighbor (k-NN) algorithms in comparison to regression analyses, and (3) to evaluate the importance of predictor variables from satellite images. Therefore, 64 sampling units were collected, and the pixel values associated with the field plots were extracted in a $3 \times 3$-pixel window surrounding the reference pixel. For multiple linear regression analyses, the $R^2$ values ranged from 0.63 to 0.78, while the $R^2$ values of the models fitted using the random forest algorithm ranged from 0.52 to 0.83 and the $R^2$ values of those fitted using the k-NN algorithm ranged from 0.30 to 0.68. The estimates made through multiple linear regression analyses showed better results for the equations adjusted for the beginning of the dry season (May and June). Adopting the random forest algorithm resulted in improvements in the statistical metrics of evaluation of the fuel load estimates for the Cerrado grassland relative to multiple linear regression analyses. The variable fraction-soil (FS) exerted the most significant effect on surface fuel load estimates, followed by the vegetation indices NDII, GVMI, DER56, NBR, and MSI, all of which use near-infrared and short-wave infrared channels in their calculations.

**Keywords:** fuel load; satellite imagery; image processing; fuel load maps; fuel estimation

## 1. Introduction

Fire is common in the Cerrado biome, mainly in open plant physiognomies predominated by herbaceous and grassland vegetation. According to Miranda et al. [1], fires in the Cerrado, like in other savannas, can be characterized as surface fires that consume the fine fuels of the grass fuel layer. Fine surface fuels represent the layer of particles less than 0.6 cm in thickness, consisting of litter, grasses, herbs, and downed woody material [2]. For Rothermel [3], fuel represents the organic matter available for ignition and combustion, and this is characterized as the only fire-related factor that can be controlled by human action. Knowing the fuel characteristics is essential for determining fire behavior and making decisions in integrated fire management and wildfire suppression activities. However, determining the characteristics of fuel is temporally and spatially complex [4,5]. Li et al. [6] reported that determining fuel characteristics demands high costs and a considerable time

for sampling. Roberts et al. [7] indicated the following essential attributes for understanding fire behavior: fuel type, fuel biomass, fuel moisture, and fuel condition (live or dead). The fuel load is a crucial variable commonly used in various fire management strategies, such as fire risk assessments and fire behavior prediction [3,8]. The fuel load has a significant influence on the modeling of biomass and carbon estimates, and its knowledge for land managers is fundamental [9]. Nevertheless, there are few studies on the Cerrado that involve determining the characteristics of its flammable material.

Remote sensing techniques are essential for estimating several fuel characteristics as more studies are conducted and improvements are made. According to Roberts et al. [7], the products obtained using various remote sensing techniques can help assess wildfire hazards, which include the following: (i) indirect measurements of live fuel moisture, (ii) measurements of live herbaceous biomass, (iii) measurements of the fuel condition, and (iv) detailed classifications of the fuel type. Van Wagtendonk and Root [10] noted that information relative to fuel is often presented as fuel model maps; fuel models are used to determine the fuel load, size, depth, and moisture of extinction. A wide range of studies have estimated fuel variables through remote sensing products. However, there is a significant lack of studies concerning the surface fuel of grassland environments of the Cerrado biome. Several studies have characterized and estimated forest biomass using Landsat imagery [11–15], MODIS products [16,17], and Lidar sensors [18,19]. Various studies that have used remote sensing products mostly relate to the characterization of forest variables, but few have worked with estimates of surface fuel variables, which are more related to the occurrence of surface fires, quite recurrent in grassland and savanna areas.

The use of vegetation indices has been widely addressed. They are used in biomass studies to determine fuel moisture [20,21]. Other studies using spectral mixture techniques have also been conducted [22,23]. However, few are related to determining surface fuel's physical characteristics, except for Franke et al. [24], who applied spectral mixture techniques to map fine fuel accumulation. The significant difference in this study is the indirect estimation of fine surface fuel classified by type into live and dead fuel and of its behavior during the dry season, differing between the beginning and end of the dry period using remote sensing products. Most studies that estimate the fuel load perform the modeling without considering different fuel types. They do not consider the seasonality between other times of the year or more critical periods and do not offer vital information for risk estimates of forest fires. Few studies consider the use of spectral mixture analysis products associated with machine learning algorithms in estimating the fuel load of grasses, mainly in environments with high spatio-temporal variability, such as the Cerrado biome. In this sense, the main objectives of this study are as follows:

(1) To evaluate the performance of multiple linear regression equations adjusted for load estimation in classes of live and dead fine fuels, considering the beginning and end of the dry season, based on the reflectance of Landsat 8 OLI images, vegetation indices, and fraction values (F-values) of the spectral mixture analysis (SMA);

(2) To assess the use of random forest and k-nearest neighbor algorithms to estimate the fine fuel load in different classes in comparison to traditional multiple linear regression analyses;

(3) To analyze the importance of each predictor variable from remote sensing products in random forest models.

## 2. Materials and Methods

### 2.1. Study Area

Collections in the field were performed in the central southern portion of the Serra Geral do Tocantins Ecological Station (EESGT), a fully protected conservation unit with an area of 716,306 ha, located in the Cerrado biome in the Jalapão region in Brazil. According to the Köppen climate classification, the region's climate is classified as Aw (tropical savanna climate), and its annual precipitation ranges from 1400 to 1500 mm [25], which is higher than the annual potential evapotranspiration. In the region, summer occurs between October

and April and is rainy and winter occurs between May and September and is dry [26]. The predominant plant physiognomy is grassland: pure grassland (campo limpo), pure wet grassland (campo limpo úmido), grassland with scattered shrubs and trees (campo sujo), wet grassland with scattered shrubs and trees (campo sujo úmido), and rocky grassland (campo rupestre) [27,28]. The predominant soil type is quartz sand or quartzipsamments, and it has a sandy or loamy texture at least 2 m deep and may have up to 15% clay. The location's relief varies from relatively flat to gently undulating, with average elevations between 300 and 500 m [29]. Figure 1 presents the study area and the spatial distribution of data sampling units.

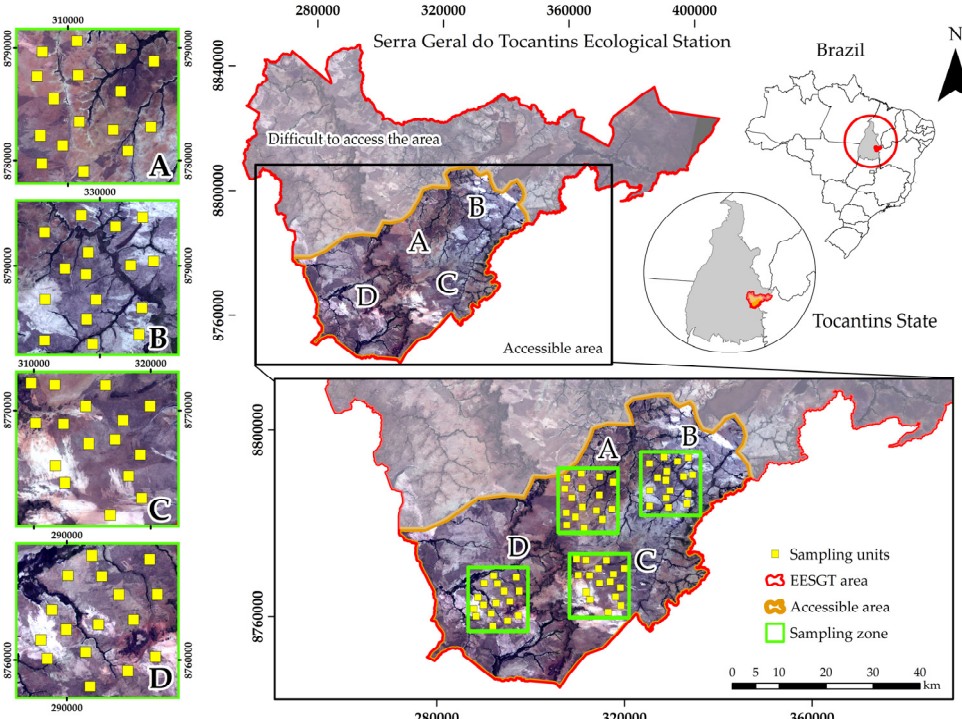

**Figure 1.** Study area and spatial distribution of data sampling units. (**A**–**D**) Fuel sampling zones with random plots distributed in the Cerrado grassland areas in the protected area: Datum SIRGAS 2000, UTM, spindle 23 South. Landsat 8 OLI images (RGB: 754), 221/67 and 221/68 (Path/Row) of date 5 May 2017.

### 2.2. Field Survey

A field survey of the variables of the surface fuel load was conducted by installing data sampling units that were collected during the periods from 11 May 2017 to 31 May 2017 (May), from 6 June 2017 to 29 June 2017 (June), from 5 August 2017 to 25 August 2017 (August), and from 8 September 2017 to 24 September 2017 (September). Average air temperatures at the time of samplings ranged from 24.4 to 31.0 °C from the beginning (May and June) to the end (August and September) of the dry season, respectively. The relative humidity of the air presented averages at the time of samplings, which varied from 41.0 to 27.3% from the beginning to the end of the dry season. Each sampling unit had two 30 m transects spaced 14 m apart. Markings were made on the transects to install, by prior draw, eight sub-samples of surface fuel with an area of 0.25 m$^2$ (0.5 × 0.5 m), and fuel was collected from them using the destructive method. The size of the sub-samples was chosen for a broader sampling that could provide a more significant variability of fuel samples in a shorter time. Accordingly, 64 data sampling units separated by a distance of over 100 m from each other were collected for analyses using satellite images. It is noteworthy that data sampling was carried out in the area of viable access, given the existence of an area of difficult access further north (Figure 1).

The destructive sampling involved separating the material according to its physiological state (live or dead) into different diameter classes (timelag), adapting the methodology proposed by Schroeder and Buck [30] and Brown et al. [2]. Thus, the surface fuels were classified as follows: (i) dead grass fuel, (ii) 1 h downed wood debris (<0.64 cm diameter), (iii) live grass fuel, and (iv) live shrub fuel (<0.64 cm diameter). As the study aimed to estimate fine fuels under 0.64 cm in diameter, which exert a more significant influence on the fire spread in regions with open vegetation in the Cerrado biome, the 10 h (0.64–2.54 cm) and 100 h (2.54–7.62 cm) classes were not considered. In addition to the abovementioned classes, the analyses and estimates took into account the total fine live (grass + shrub < 0.64 cm diameter) and fine dead (grass + 1 h downed wood debris) fuels and the total fine fuel load (live and dead; <0.64 cm diameter). After collecting the sub-samples from the field, they were all dried until they reached a constant mass. Next, the dry mass was determined, and its respective fuel moisture content was calculated using the ratio between wet mass and dry mass on a dry-weight basis and then converted to a percentage. Figure 2 presents a diagram exemplifying the collection locations, surface fuel data sampling, and characterization process.

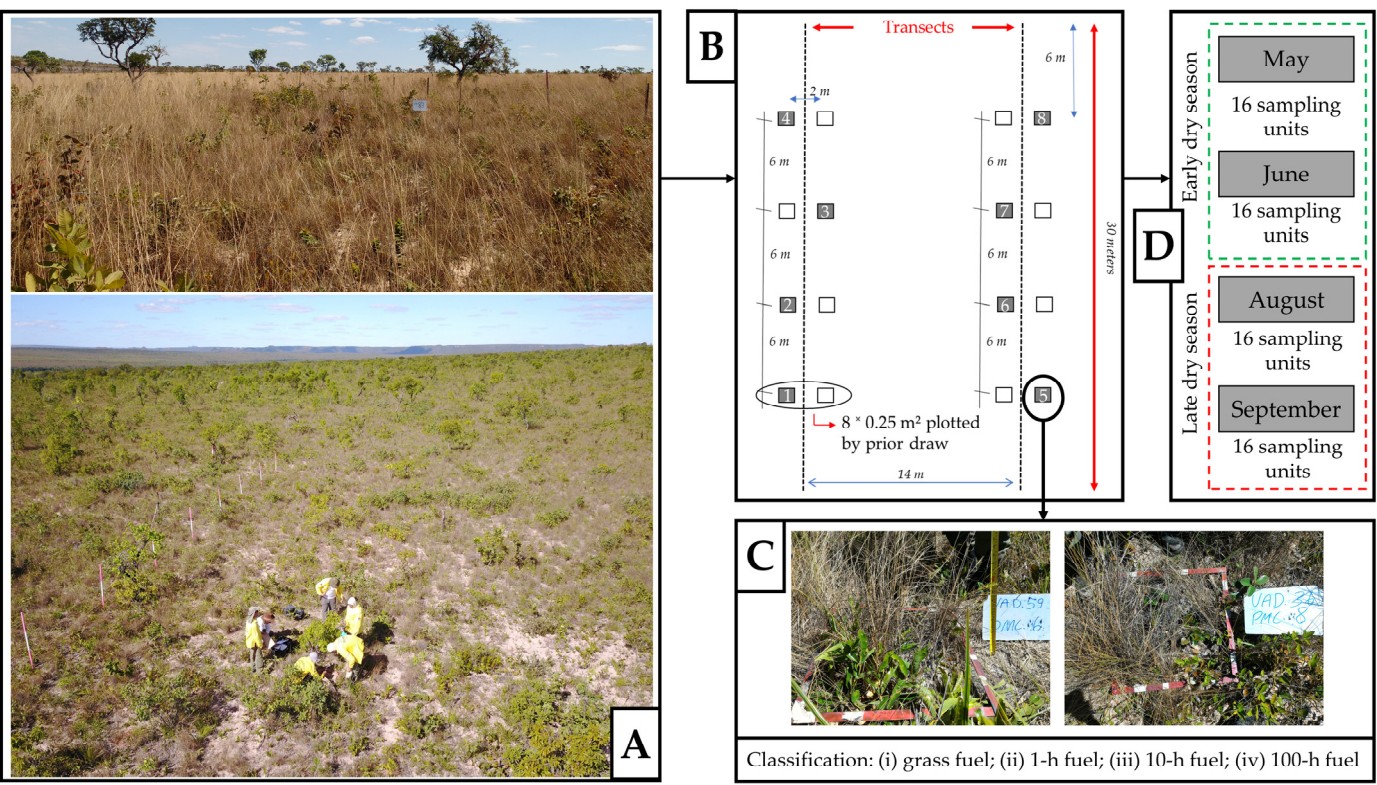

**Figure 2.** Overview of fuel sampling. (**A**) Physiognomy of some sampling areas, (**B**) fuel data sampling unit, (**C**) sub-samples of surface fuel characterization, and (**D**) number of samples taken at the beginning (May and June) and end of the dry season (August and September).

### 2.3. Obtaining and Processing Satellite Imagery

Images from the Landsat 8 OLI satellite were used in this study. They were obtained for the approximate dates of four collections of field samples during the dry season in May and June (beginning of the dry season) and in August and September (end of the dry season), corresponding to the following dates (dd month yyyy): 5 May 2017, 6 June 2017, 22 June 2017, 24 July 2017, 9 August 2017, and 10 September 2017. To cover the entire study area, images from 3 Landsat 8 OLI scenes were used, namely: 220/067, 221/067, and 221/068. The reflectance values of the bands were extracted, and the vegetation indices were calculated through the cloud-based geospatial data processing platform Google Earth Engine (GEE). The GEE platform provides easy access to high-performance computing resources

for large-scale processing of geospatial data sets [31]. Landsat 8 OLI Surface Reflectance data were generated using the Land Surface Reflectance Code (LaSRC) algorithm [32]. The reflectance values were obtained from the Landsat 8 OLI sensor bands in the following channels: (i) blue (0.45–0.51 μm), (ii) green (0.53–0.59 μm), (iii) red (0.64–0.67 μm), (iv) near-infrared (0.85–0.88 μm), (v) short-wave infrared (1.57–1.65 μm), and (vi) short-wave infrared (2.11–2.29 μm).

### 2.3.1. Vegetation Indices (VIs)

Based on the combination of reflectance in different spectral bands, vegetation indices (VIs) were calculated. VIs are an efficient way to attain empirical information from multispectral sensors and are frequently used in various scientific studies that set out to better explain or determine the behavior of certain vegetation variables [10,11,33,34]. Therefore, a total of 20 vegetation indices were calculated and incorporated into the analyses as independent variables: (i) Normalized Difference Vegetation Index (NDVI) [35], (ii) Visible Atmospherically Resistant Index (VARI) [36], (iii) Visible Green Index (VIgreen) [36], (iv) Simple Ratio (SR) [37], (v) Structure Insensitive Pigment Index (SIPI) [38], (vi) Soil-Adjusted Vegetation Index (SAVI) [39], (vii) Normalized Difference Water Index (NDWI) [40], (viii) Normalized Difference Infrared Index (NDII6) [41], (ix) Normalized Burn Ratio (NBR) [41], (x) Modified Vegetation Index (MVI) [42], (xi) Modified Simple Ratio (MSR) [43], (xii) Moisture Stress Index (MSI) [44], (xiii) Modified Normalized Difference Water Index (MNDWI) [45], (xiv) Integral (INT) [46], (xv) Global Vegetation Moisture Index (GVMI) [33], (xvi) Enhanced Vegetation Index (EVI) [47], (xvii) Derivative-band 5 (ivp), band 6 (siwir1) (DER56) [48], (xviii) Derivative-band 4 (vm), band 5 (ivp) (DER45) [48], (xix) Derivative-band 3 (vd), band 4 (vm) (DER34) [48], and (xx) Derivative-band 2 (az), band 3 (vd) (DER23) [48]. The vegetation indices analyzed in this study show relationships with the fuel load and moisture, as well as with the physiological state of the fuel. These vegetation indices were chosen with the aim to analyze their viability for estimating the fuel load from Landsat 8 OLI images, considering seasonality during the dry season in a Cerrado environment.

### 2.3.2. Spectral Mixture Analysis (SMA)

The spatial variability of vegetation within a pixel is a strong limiting factor in ascertaining the fuel condition and load information. Spectral mixture analysis (SMA) has considerable potential to estimate the fuel condition and fuel moisture content at the sub-pixel level [34]. SMA is a widely applied technique in remote sensing for obtaining ecologically relevant and significant components of an image pixel [49]. In SMA, two or more reference spectra (endmembers), such as green vegetation, dry vegetation, soil, and shade, are modeled as linear combinations to estimate the sub-pixel fractions of each component [50,51].

After retrieval, the Landsat 8 OLI images underwent the atmospheric correction process to eliminate interferences from the atmosphere to extract data and determine reflectance values [32,52]. Next, color composites were obtained and used in subsequent study steps. The color composite of the Landsat 8 OLI image used in the initial collection for the spectral mixture analyses was 7 (short-wave infrared-SWIR 2), 5 (near-infrared), and 4 (red). After processing the satellite images, the creation of a spectral library began. First, the surface targets for SMA were divided into (i) non-photosynthetic dry vegetation (NPV), (ii) green vegetation (GV), and (iii) exposed soil. This classification was chosen because green vegetation, dry vegetation, and soil dominate Cerrado grassland environments. Several candidates for "pure pixels," known as endmembers, were selected from each target. Thus, based on the specific spectral behavior of each target, regions of interest (ROI) whose reflectance presented spectral behaviors that were as faithful as possible to the characteristic spectral curve of the respective target were delimited.

To fulfill the objectives proposed herein, endmembers were chosen in four representative images of the four months of the dry season (May, June, August, and September)

according to the dates of the Landsat 8 images previously mentioned. After the selection of all the pure pixel candidates of the features, as mentioned earlier, based on the spectral curves, the process of selecting endmembers based on the lowest statistical errors began. The multiple endmember spectral mixture analysis (MESMA) methodology was applied to perform this procedure [53] using MESMA QGIS Plugin (version 1.0.8) [54]. MESMA's results produce grayscale images whose values (called fraction values (F-values)) represent the estimated relative degree in which each pixel corresponds to its respective reference spectrum. Therefore, the images generated with F-values were for the GV, NPV, and soil components.

### 2.3.3. Data Extraction

The pixel values associated with the field plots were extracted in a 3 × 3-pixel window around the reference pixel (pixel associated with the field plot). This way, the median values of the pixels in each 3 × 3-pixel polygon related to the field plots were extracted [11,55–59]. The data were exported and tabulated to compare the information collected from the plots in the field with the information from the spectral mixture analysis, reflectance, and calculated vegetation indices.

### *2.4. Statistical Analysis of the Data*
### 2.4.1. Multiple Linear Regression Analysis

After processing the data from the Landsat 8 OLI satellite images, all the information associated with the collection points in the field were extracted in a 3 × 3-pixel window. Next, all the data were tabulated and correlated with the fuel load values in the different classes. In multiple linear regression analyses, the fuel load was a dependent variable, especially for the fine dead fuels (dead grass fuel, 1 h downed wood debris, and total fine dead fuel) and total fine fuel (live and dead, <0.64 cm diameter) classes. The predictor (independent) variables were divided into three main groups: (i) fraction values (F-values) from the spectral mixture analysis (SMA), (ii) reflectance of the Landsat 8 OLI images in different spectral bands, and (iii) the calculated vegetation indices. The stepwise method was adopted to select the most relevant variables for the regression equations. Also, the adjustments of multiple linear regression equations considered the following: (i) adjusting equations considering 32 plots collected in the initial months of the dry season (May and June), (ii) adjusting equations considering 32 plots collected in the final months of the dry season (August and September), and (iii) adjusting equations for the entire period of the dry season (all 64 plots). Figure 3 presents a flowchart of the methodology applied in this study.

### 2.4.2. Random Forest Machine Learning Algorithm

The random forest (RF) machine learning algorithm is a non-parametric method developed by Breiman [60] that has been widely adopted in various studies [12,16,58,61]. The RF algorithm combines several predictor trees generated by an independently sampled random vector with a similar distribution for all trees in the forest [62]. Compared to traditional linear regression methods, random forest models exhibit several advantages in above-ground biomass estimation, such as better accuracy and fewer non-linear problems [63].

Although the training and test sets were divided randomly in R, a specific code command (set seed) was used to guarantee the model's reproducibility—in other words, to run the model with the same previously defined data sets (training and test). To implement a random forest, two parameters must be defined: the number of trees (ntree) and the number of features in each split (mtry). Regarding ntree, in this study, it was observed that the prediction accuracy with the random forest algorithm did not increase significantly when ntree presented values greater than 1000. The ideal ntree value defined was ntree = 1000 to guarantee the reliability of the prediction results without affecting computational efficiency [64,65]. For mtry, the values were defined as one-third of the

total number of predictor variables [60,66–68]. These varied between mtry = 4 and 5 for models adjusted with predictor variables selected using the stepwise method and mtry = 10 for models adjusted with all predictor variables. The "randomForest" package in the R program was used to apply the algorithm. The importance of the predictor variables used in the random forest models was evaluated with the increase in the mean-square error (% IncMSE), which measures the effect of a variable's predictive potential when subjected to a random permutation. In short, this metric shows how much the model's accuracy decreases if a particular predictor variable is left out of the model.

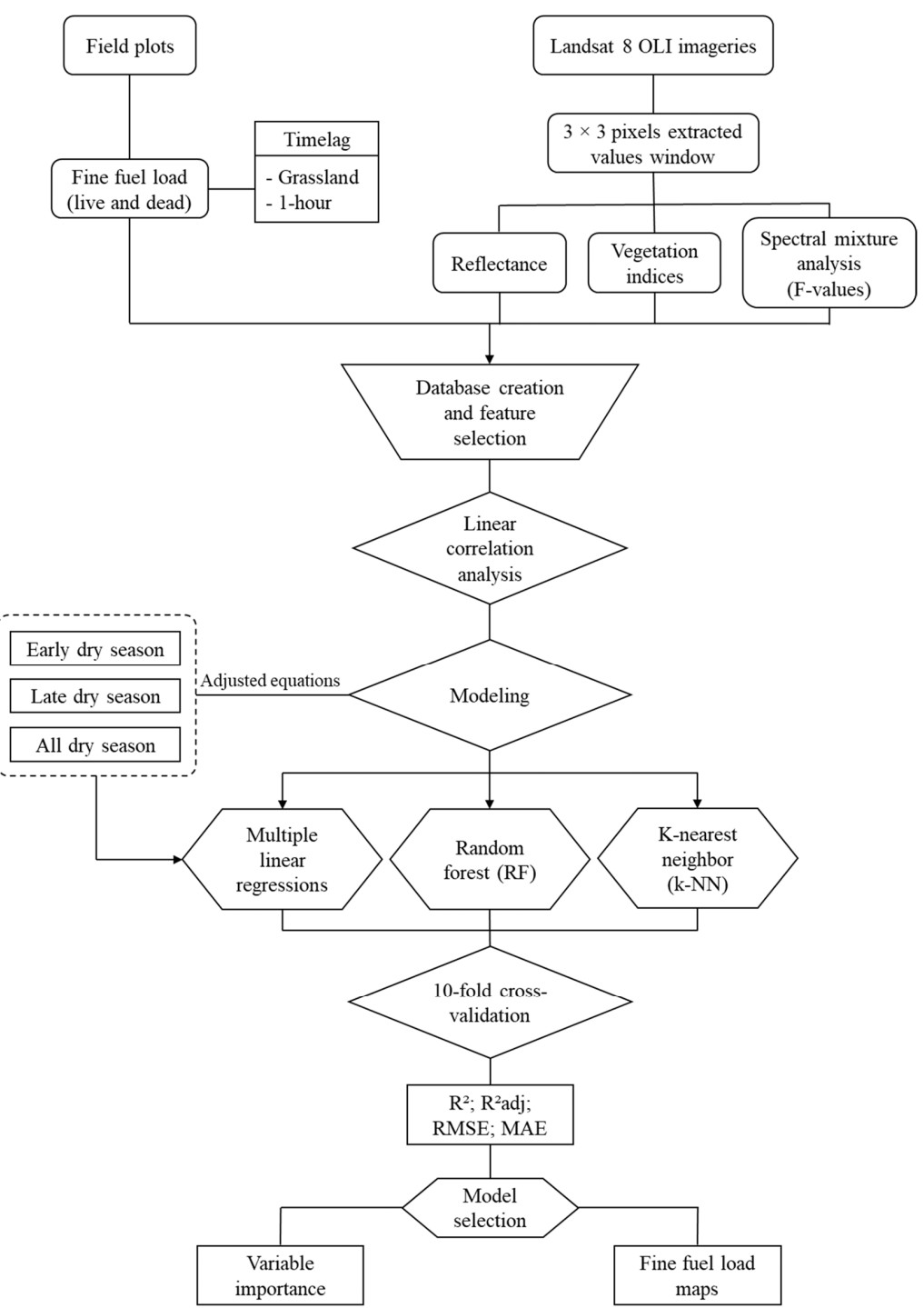

**Figure 3.** Flowchart of the methodology applied in this study. $R^2$ = coefficient of determination, $R^2$adj = adjusted coefficient of determination, RMSE = root-mean-square error, MAE = mean absolute error.

### 2.4.3. K-Nearest Neighbor Machine Learning Algorithm

Another machine learning algorithm used to verify the feasibility of estimating the fuel load from satellite imagery products was the k-nearest neighbor (k-NN) algorithm. The k-NN algorithm is used for classification and regression. It predicts the values of specific variables based on the similarity in a covariate space between the point and other points representing the values of the variables [69]. The use of the k-NN algorithm associated with remote sensing data has played a key role in forest studies, and there have been several applications in forest inventories in the field that aimed to improve the accuracy of estimates [70,71]. The k-NN algorithm was applied using the "Caret" package in R. There is one essential parameter in this algorithm, K, or the number of neighbors that need to be optimized. This parameter specifies the voting system in the k-NN algorithm wherein the K nearest neighbors are selected, and then it sets the output model [72]. The K values were determined based on the lowest RMSE values, and the optimal values were chosen for each estimated fuel variable.

### 2.4.4. Cross-Validation

Cross-validation evaluates a model's performance and provides a nearly unbiased prediction error estimator [73]. The multiple linear regression equations, random forest models, and k-NN algorithm models were subjected to cross-validation with the k-fold cross-validation technique using 10 folds. The k-fold cross-validation technique is a robust method for estimating a model's accuracy, and it generally provides more accurate estimates of the test error rate [11,74,75]. The following statistical metrics were used to validate the equations: (i) the values of the squared coefficient of determination ($R^2$), (ii) the adjusted squared coefficient of determination ($R^2$adj), (iii) the root-mean-square error (RMSE), and (iv) the mean absolute error (MAE). The validations considered the adjustments of the multiple linear regression models and the random forest and k-nearest neighbor algorithm models, with an emphasis on the models with the best fit considering all the independent variables of the study and on the models with the best fit considering the independent variables selected using the stepwise method.

## 3. Results

### 3.1. Multiple Linear Regression Analysis

Table 1 presents the metrics of the multiple linear regression equations, which were validated by considering the study's independent variables. The multiple linear regression equations performed best with adjustments using the stepwise method. Table 1 presents only the metrics of the equations adjusted using this method. The equations validated using the k-fold cross-validation method, which were notable for the coefficient of determination ($R^2$), were for the estimation of the total fine dead fuel (grass + 1 h downed wood debris), with $R^2 = 0.92$, and dead grass fuel, with $R^2 = 0.89$. These equations mentioned earlier are for the beginning of the dry season ($p < 0.05$). However, the lowest RMSE and MAE values were observed for estimates of dead grass fuel at the beginning of the dry season, with values below 0.40. The statistical metrics verified for the end of the dry season were lower than those observed for the equations adjusted for the beginning of the dry season in the study area.

For the entire period of the dry season, the $R^2$ values ranged from 0.57 to 0.78. The equation for estimating dead grass fuel performed the best, with $R^2 = 0.78$, followed by the stepwise equation for calculating the total fine dead fuel (grass + 1 h downed wood debris), with $R^2 = 0.73$ ($p < 0.05$). The $R^2$adj values of the equations subjected to cross-validation may be negative without a lower limit and present a maximum value of 1, as noted by Peterson et al. [76], which was the case for the equations at the end of the dry season. The lowest RMSE and MAE values were found to estimate dead grass fuel.

**Table 1.** Statistical metrics of the cross-validation of multiple linear regression models with the best fit for fuel load estimation.

| Per | Estimated Fuel (y) | Predictor Variables (x) | $R^2$ | $R^2$adj | RMSE | MAE |
|---|---|---|---|---|---|---|
| 1 | Dead grass | NPV; soil; B5: nir; B6: swir 1; B7: swir 2; VARI; SR; SIPI; NDII; MSR; MSI; GVMI; DER56; DER34 | 0.89 | 0.81 | 0.37 | 0.32 |
| | 1 h Downed wood debris | NPV; soil; GV; B4: red; B5: nir; B6: swir 1; B7: swir 2; NDVI; VARI; VIgreen; SR; SIPI; SAVI; NDWI; NRR; MSI; MNDWI; DER34; DER23 | 0.81 | 0.55 | 0.45 | 0.37 |
| | Total fine dead (grass + 1 h downed wood debris) | NPV; soil; GV; B4: red; B5: nir; B6: swir 1; B7: swir 2; VARI; VIgreen; SR; SIPI; SAVI; NDWI; NDII6; NBR; DER45 | 0.92 | 0.84 | 0.49 | 0.40 |
| | Total fine (live and dead) | NPV; soil; B2: blue; B4: red; B5: nir; B6: swir 1; NDVI; VARI; VIgreen; SR; SIPI; SAVI; NDWI; NDII6; NBR; MSR; MSI; MNDWI; INTEGRAL; DER45; DER34; DER23; GVMI | 0.81 | 0.37 | 1.26 | 1.06 |
| 2 | Dead grass | NPV; Soil; B2: blue; B3: green; B5: nir; B6: swir 1; B7: swir 2; NDVI; VIgreen; SR; NDWI; NDII6; NBR; MSR; MSI; MNDWI; GVMI; EVI; DER56; DER34; DER23 | 0.81 | 0.47 | 0.44 | 0.38 |
| | 1 h Downed wood debris | NPV; soil; GV; B2: blue; B3: green; B4: red; B5: nir; B7: swir 2; NDVI; VARI; VIgreen; SR; SIPI; SAVI; NDWI; NDII6; NRR; MSR; MSI; INTEGRAL; GVMI; EVI; DER56; DER45; DER34; DER23 | 0.45 | −1.59 | 1.90 | 1.53 |
| | Total fine dead (grass + 1 h downed wood debris) | Soil; GV; B2: blue; B3: green; B4: red; B5: nir; B6: swir 1; B7: swir 2; NDVI; VARI; VIgreen; SR; SIPI; SAVI; NDWI; NDII6; NBR; MSR; MSI; INTEGRAL; GVMI; EVI; DER56; DER45; DER34; DER23 | 0.64 | −0.72 | 2.24 | 1.98 |
| | Total fine (live and dead) | Soil; GV; B2: blue; B3: green; B4: red; B5: nir; B6: swir 1; B7: swir 2; NDVI; VARI; VIgreen; SR; SAVI; NDWI; NDII6; MSR; MSI; INTEGRAL; EVI; DER56; DER45; DER34; DER23 | 0.56 | −0.44 | 2.27 | 2.01 |
| 3 | Dead grass | Soil; B3: green; B4: red; B5: nir; B6: swir 1; VARI; VIgreen; SR; SAVI; NDWI; NDII6; MSR; INTEGRAL; GVMI; EVI; DER56 | 0.78 | 0.72 | 0.36 | 0.31 |
| | 1 h Downed wood debris | NPV; soil; B2: blue; B4: red; B7: swir 2; VARI; SR; SIPI; NDWI; NRR; MSI; INTEGRAL; GVMI; EVI; DER45 | 0.57 | 0.45 | 0.62 | 0.51 |
| | Total fine dead (grass + 1 h downed wood debris) | NPV; soil; B7: swir 2; VARI; VIgreen; SIPI; NBR; MSR; MSI; INTEGRAL; GVMI; EVI; DER45 | 0.73 | 0.66 | 0.75 | 0.62 |
| | Total fine (live and dead) | NPV; soil; GV; B2: blue; B4: red; B5: nir; B6: swir 1; B7: swir 2; VARI; SIPI; NDWI; NBR; DER56; DER34 | 0.63 | 0.54 | 1.06 | 0.91 |

Per = dry season period: (1) early dry season (May and June), (2) late dry season (August and September), and (3) all dry season (May to September).

### 3.2. Estimates Using the Random Forest Algorithm

### 3.2.1. Model Performance Metrics

To use the random forest algorithm, the entire dry season was considered for the adjustment of the models. The models were not divided for the dry season's beginning and end, as was performed in multiple linear regression analyses. Like the regression equations, the random forest algorithm was adjusted for each estimated variable by considering all the study's independent variables and the variables selected using the stepwise method. Unlike the multiple linear regression equations, the random forest models presented a slight advantage in the adjustments with all the independent variables compared to the adjustments with the variables selected using the stepwise method, as shown in Table 2. The $R^2$ values with the random forest algorithm ranged from 0.52 to 0.83; the highest values were found to estimate dead grass fuel and the total fine dead fuel.

**Table 2.** Statistical metrics used to assess the models generated using the random forest algorithm.

| Estimated Fuel (y) | Predictor Variables (x) | $R^2$ | $R^2$adj | RMSE | MAE |
|---|---|---|---|---|---|
| Dead grass | All predictor variables | 0.83 | 0.71 | 0.33 | 0.24 |
| | * Soil; B3: green; B4: red; B5: nir; B6: swir 1; VARI; VIgreen; SR; SAVI; NDWI; NDII6; MSR; INTEGRAL; GVMI; EVI; DER56 | 0.83 | 0.78 | 0.33 | 0.23 |
| 1 h Downed wood debris | All predictor variables | 0.59 | 0.30 | 0.58 | 0.44 |
| | * NPV; soil; B2: blue; B4: red; B7: swir 2; VARI; SR; SIPI; NDWI; NRR; MSI; INTEGRAL; GVMI; EVI; DER45 | 0.52 | 0.38 | 0.61 | 0.46 |
| Total fine dead (grass + 1 h downed wood debris) | All predictor variables | 0.83 | 0.71 | 0.59 | 0.44 |
| | * NPV; soil; B7: swir 2; VARI; VIgreen; SIPI; NBR; MSR; MSI; INTEGRAL; GVMI; EVI; DER45 | 0.79 | 0.74 | 0.63 | 0.49 |
| Total fine (live and dead) | All predictor variables | 0.62 | 0.35 | 0.89 | 0.75 |
| | * NPV; soil; GV; B2: blue; B4: red; B5: nir; B6: swir 1; B7: swir 2; VARI; SIPI; NDWI; NBR; DER56; DER34 | 0.55 | 0.43 | 0.96 | 0.81 |

* Variables selected using the stepwise method in multiple linear regression analyses.

To estimate the total fine fuel (live and dead), the best performance was related to the model using all the independent variables, with $R^2 = 0.62$. For the RMSE and MAE values of the estimates using the random forest algorithm, the lowest values were recorded for the estimation of dead grass fuel, as 0.33 and 0.23, respectively. In contrast, the highest RMSE and MAE values corresponded to the estimation of the total fine fuel load (live and dead). Figures 4 and 5 show the fine fuel load mappings in the Serra Geral do Tocantins Ecological Station, with the best fits for each fuel load category analyzed.

### 3.2.2. Importance of the Variables in the Random Forest Models

In analyzing the results of the importance of the variables in the RF models presented in Figure 6, the considerable importance of the fraction-soil (FS; F-values) variable from the spectral mixture analysis (SMA) was noteworthy in all the models addressed. In all situations, the FS variable increased the mean-square error (% IncMSE) above 20% in all the models presented. This means that the removal of the predictor FS variable resulted in an increase in the mean-square error above 20% for the estimated fuel variables. Among the fuel variables in which FS exerted a greater effect, the total fine dead fuel (grass + 1 h downed wood debris) and dead grass fuel were notable, presenting values of 33.5% and 29.1%, respectively.

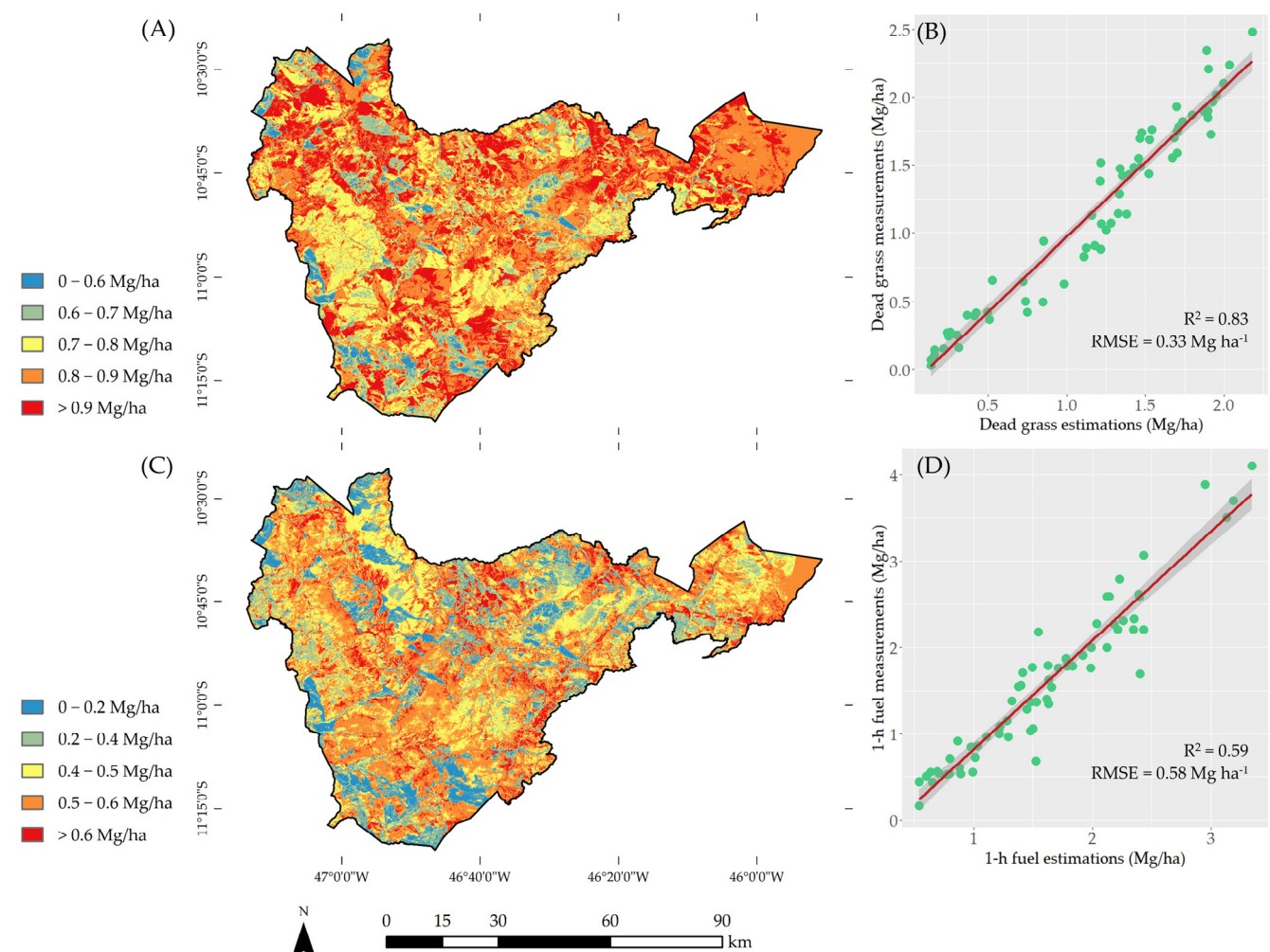

**Figure 4.** Fine fuel load mappings performed using the best RF models: (**A**,**B**) estimates for dead grass fuel and (**C**,**D**) estimates for 1 h downed wood debris.

The vegetation indices NDII, NBR, GVMI, and DER56 emerged as the second-most essential variables considering the estimates of dead grass fuel, 1 h downed wood debris, the total fine dead fuel (grass + 1 h downed wood debris), and the total fine fuel (live and dead), respectively. The GVMI appeared most frequently among the vegetation indices and was most important in the fitted random forest models. It was not a variable of importance only for estimating the total fine fuel. Out of the bands' reflectance, the short-wave infrared channel (SWIR2-2.11–2.29 μm) occurred most frequently and had the most significant importance. Figure 6 graphically illustrates the importance of the random forest algorithm models with the best fit.

### 3.3. Estimates using the K-Nearest Neighbor Algorithm

Similarly, the data from the same database used for the application of the RF algorithm were executed with the k-nearest neighbor (k-NN) algorithm so that two models were built for each fuel load variable: one model considering all the independent variables of the study and another model considering only the variables selected using the stepwise method. As a parameter for using the k-NN algorithm, the K values were chosen based on the lowest RMSE values, ranging from 5 to 13. Table 3 demonstrates the K values chosen for each k-nearest neighbor algorithm model.

(A)

0 – 1.0 Mg/ha
1.0 – 1.4 Mg/ha
1.4 – 1.6 Mg/ha
1.6 – 1.8 Mg/ha
> 1.8 Mg/ha

(B)

R² = 0.83
RMSE = 0.59 Mg ha⁻¹

(C)

0 – 3.5 Mg/ha
3.5 – 4.0 Mg/ha
4.0 – 4.5 Mg/ha
4.5 – 4.8 Mg/ha
> 4.8 Mg/ha

(D)

R² = 0.62
RMSE = 0.89 Mg ha⁻¹

**Figure 5.** Fine fuel load mappings were performed using the best RF models: (**A**,**B**) estimates for the total fine dead fuel and (**C**,**D**) estimates for the total fine fuel.

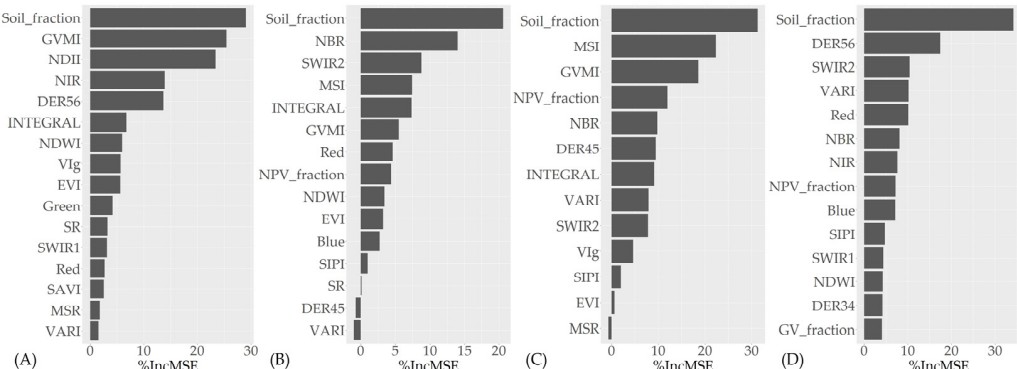

**Figure 6.** Importance of the variables of the random forest (RF) algorithm models with the best fit for fuel load estimation. (**A**) RF model for dead grass fuel, (**B**) RF models for 1 h downed wood debris, (**C**) RF models for the total fine dead fuel, and (**D**) RF models for the total fine fuel. %IncMSE indicates the increase in the mean-square error as a percentage.

**Table 3.** K values chosen for each k-nearest neighbor algorithm model.

| Estimated Fuel (y) | Predictor Variables (x) | RMSE Value | Chosen K Value |
|---|---|---|---|
| Dead grass | All predictor variables | 0.4072 | 5 |
| | Stepwise * | 0.4453 | 7 |
| 1 h Downed wood debris | All predictor variables | 0.8183 | 7 |
| | Stepwise * | 0.8253 | 9 |
| Total fine dead (grass + 1 h downed wood debris) | All predictor variables | 1.1051 | 5 |
| | Stepwise * | 1.1233 | 9 |
| Total fine (live and dead) | All predictor variables | 1.6060 | 7 |
| | Stepwise * | 1.6136 | 13 |

* Variables selected using the stepwise method in multiple linear regression analyses.

Like the random forest models, the k-nearest neighbor models presented slightly superior performances for the adjustments made with all the predictor variables of the study in comparison with those selected using the stepwise method. The $R^2$ values ranged from 0.30 to 0.68, and the k-NN models with the best fit were those for estimating dead grass fuel, followed by the model for estimating the total fine dead fuel, with $R^2 = 0.54$. Table 4 provides the statistical metrics of the k-NN algorithm models.

**Table 4.** Statistical metrics used to assess the models generated using the k-NN algorithm.

| Estimated Fuel (y) | Predictor Variables (x) [1] | $R^2$ | $R^2$adj | RMSE | MAE |
|---|---|---|---|---|---|
| Dead grass | All predictor variables | 0.68 | 0.45 | 0.37 | 0.30 |
| | Stepwise * | 0.61 | 0.49 | 0.41 | 0.34 |
| 1 h Downed wood debris | All predictor variables | 0.34 | −0.13 | 0.74 | 0.61 |
| | Stepwise * | 0.31 | 0.11 | 0.76 | 0.63 |
| Total fine dead (grass + 1 h downed wood debris) | All predictor variables | 0.54 | 0.21 | 0.92 | 0.78 |
| | Stepwise * | 0.49 | 0.37 | 0.98 | 0.84 |
| Total fine (live and dead) | All predictor variables | 0.38 | −0.07 | 1.43 | 1.23 |
| | Stepwise * | 0.30 | 0.12 | 1.53 | 1.31 |

* Variables selected using the stepwise method in multiple linear regression analyses. [1] The same predictor variables are shown in Table 2 for RF models.

Regarding the RMSE and MAE values, the estimates using the k-NN algorithm presented the lowest values for the estimation of dead grass fuel, as 0.37 and 0.30, respectively. The highest RMSE and MAE values with the k-NN models were found to estimate the total fine fuel load (live and dead) (Table 4).

## 4. Discussion

Given the low correlations of the live fuel classes (live grass and live shrub), statistical analyses for their estimation could not be conducted. However, according to the findings of Santos et al. [77], the low moisture level of Cerrado grassland species is the main factor behind the increase in the fuel's flammability. Moreover, the proportion of fine dead fuel to the proportion of live fuel during the dry season in the biome can reach 70%, or over 80%, considering only the grass fuel of the area, based on the fuel characterization by Santos et al. [78]. In addition, the fine thickness of 1 h downed wood debris and grass fuels in a physiologically inactive state (e.g., grass and 1 h downed wood debris) makes the moisture content of these fuels more sensitive to changes in atmospheric conditions, thus affecting their ignition capacity and leaving areas with a predominance of these types of fuels more prone to the occurrence of wildfires, especially in the dry season [79]. As reported by Slijepcevic et al. [80] and Soares et al. [81], low fuel moisture is one of the main factors contributing to the occurrence of large fires.

Out of the adjustments of the multiple linear regression equations for the beginning and end of the dry season in the Cerrado biome, the equations adjusted for the beginning of the dry season had the best statistical performance. This may be explained by analyzing the results obtained by Santos et al. [78], who characterized the fuel of Cerrado grassland vegetation. In this study, fuel characteristics, such as the number of individuals, the number of species, the grass height, and the fuel load in different classes (timelag), exhibited different behaviors between the first and last months of the dry season in the physiognomy under analysis. The fuel condition mapping showing fractions of the NPV, GV, and soil for the beginning of the dry season (May) in contrast to the fuel condition at the end of the dry season (September) is presented in Figures 7A and 7B, respectively. For example, given the different ages of the fuels present in the area, the load values of dead grass fuel in the first months of the dry season (May and June) showed a statistically significant difference in dead grass fuel between the ages of one and four years. By the end of the dry season (August and September), no statistically significant difference was found among the fuel loads of dead grass fuels between the ages of two and four years ($p < 0.05$).

Considering all the adjustments of the multiple linear regression equations in the study (beginning, end, and the entire dry period), the $R^2$ values ranged from 0.45 to 0.92. For the whole of the dry period, the highest value was 0.78 for the adjusted equation to estimate dead grass fuel (Table 1). Similar results were reported by Tucker et al. [82], who presented more expressive values, considering only the coefficient of determination ($R^2$) values and the 0.385 μm range of the electromagnetic spectrum as the predictor variable to estimate the total dry biomass ($R^2 = 0.80$). Using Landsat 8 OLI images for fuel load estimation in the Hulunbuir grassland, Bao et al. [14] obtained an $R^2$ value of 0.64 using models based on multiple linear regression. Using Landsat 5 TM imagery to estimate the above-ground biomass in interior Alaska, Ji et al. [11] obtained an $R^2$ value of 0.73 in their regression model. In one of the few studies conducted in the Cerrado, Franke et al. [24] found relationships between variables obtained from spectral mixture analyses and the biomass of fine surface fuels in their regression model, with $R^2$ values of 0.81 (fraction of non-photosynthetic dry vegetation (FNPV)) and 0.65 (fraction-soil (FS)). The results of Franke et al. [24] differ from those of this study: between the variables given by the spectral mixture analysis and fuel load, the most substantial relationships were found for the fraction-soil (FS) values. However, the linear regression models fitted by Franke et al. [24] did not consider a cross-validation analysis. It is possible to note that there are few studies on estimating surface fuel from satellite imagery, especially for the Cerrado biome.

Like the adjustments of the multiple linear regression equations, the models resulting from applying the random forest algorithm with better performance based on the evaluation metrics were for estimating dead grass fuel classes and the total fine fuel class (grass + 1 h downed wood debris). They presented higher values than the other classes, reaching $R^2 = 0.83$. The statistical metrics obtained for estimating the 1 h downed wood debris fuel load demonstrated poor performance in relation to the other variables (Table 2). This may be explained by less spatial continuity compared to the dead grass fuel present in the study area. Dube and Mutanga [83] used Landsat 8 OLI and 7 ETM+ images and the reflectance of the bands and vegetation indices as predictor variables. They tested the random forest algorithm to estimate the above-ground biomass in southern Africa and found coefficient of determination ($R^2$) values ranging from 0.43 to 0.65. Pierce et al. [84] used the RF algorithm and information from Landsat 5 TM, field data, and topographic factors for modeling and mapping canopy fuels in California (USA) and attained pseudo-$R^2$ values ranging from 0.55 to 0.68. Frazier et al. [12] characterized the above-ground biomass in a boreal forest using Landsat temporal segmentation metrics and found $R^2$ values of 0.62 estimated using random forest models. Gao et al. [16] found higher $R^2$ values (0.75) using MODIS sensor data to estimate the above-ground biomass in an Asian region.

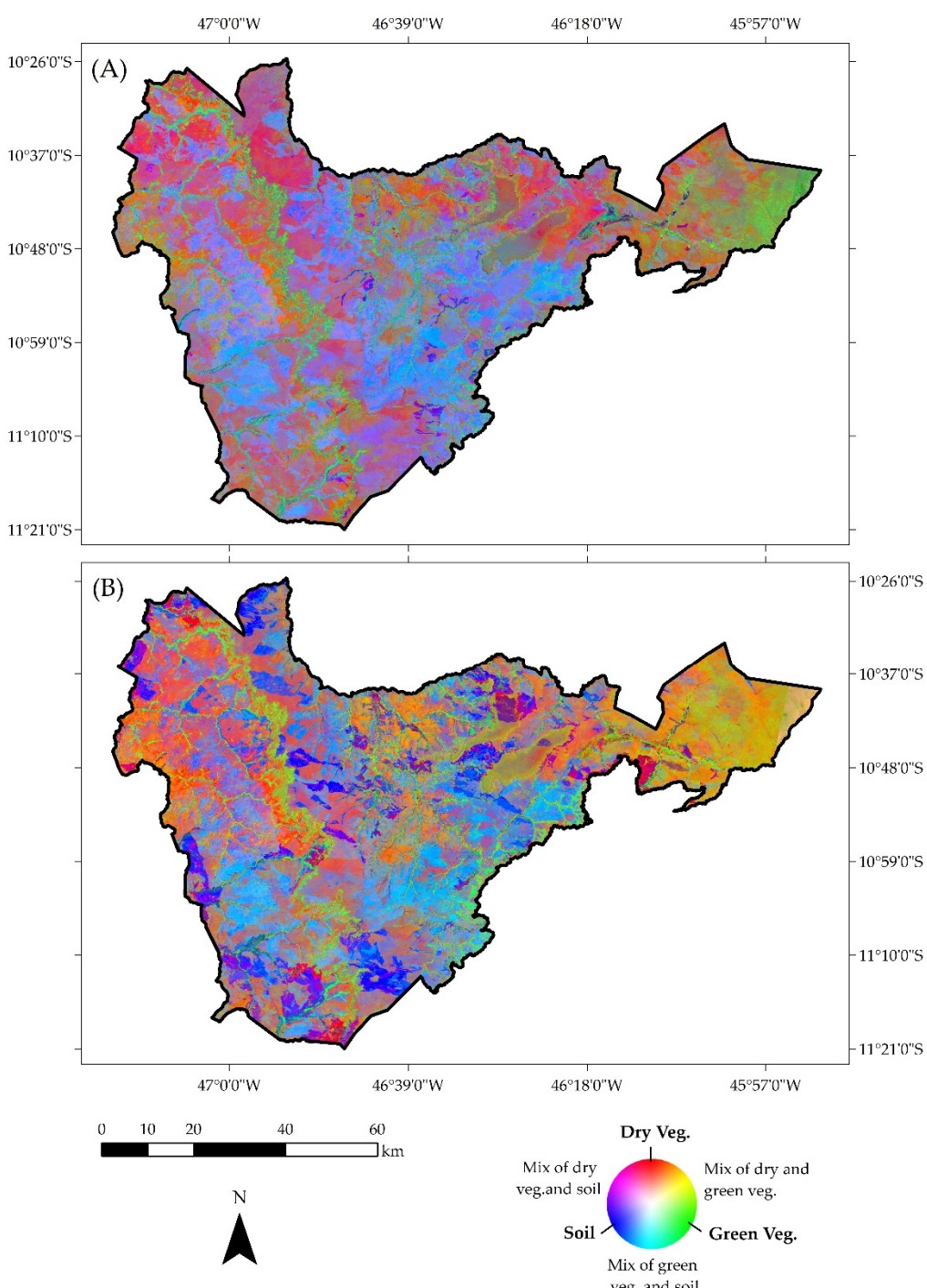

**Figure 7.** Fuel condition map consisting of the three sub-pixel fraction images (R: NPV; G: GV; B: soil) in the Serra Geral do Tocantins Ecological Station for (**A**) the beginning (May 2017) and (**B**) the end of the dry season (September 2017).

As for the importance of the variables of the RF models with the best fit, for the dead fuel classes (dead grass, 1 h downed wood debris, and total fine dead fuels) and the total fine fuel (live and dead), the fraction-soil (FS) variable from the spectral mixture analysis (SMA) was the independent variable that exerted the most significant effect. These results corroborate the higher correlations between the FS variable and the variables obtained from the spectral mixture analysis. The physiognomy of the Cerrado grassland is predominantly open (pure grassland and grassland with scattered shrubs and trees) and has a sparse population of tree species. Given the age of the fuel and its level of cover, greater or less soil

exposure becomes quite noticeable in the response to the spectral mixture analysis, which demonstrated an inverse relationship with the surface fuel load. Despite the importance of the FS variable in the RF models, there is no knowledge of its performance in areas with a greater presence of tree species. At least five vegetation indices, namely the NDII, GVMI, DER56, NBR, and MSI, had a degree of importance concerning %IncMSE above 10%. All these vegetation indices revealed the presence of the near-infrared (NIR: 0.85–0.88 μm) and short-wave infrared (SWIR1: 1.57–1.65 μm; SWIR2: 2.11–2.29 μm) channels in their respective calculations. This indicates the importance of the near-infrared and short-wave infrared channels for load estimates using both RF models and multiple linear regression analyses. The near-infrared and short-wave infrared channels have been used to estimate vegetation characteristics in several settings [55,76].

Overall, the statistical metrics used to assess the models were superior for the RF models in relation to the multiple linear regression equations and k-NN models. Thus, the models estimating the surface fuel of the Cerrado grassland using the random forest algorithm with data retrieved from satellite images provided better estimates of the surface fuel. Only the total fine fuel variable (live and dead) estimated using multiple linear regression analysis resulted in a slightly higher coefficient of determination ($R^2$), considering the entire dry period in the study area. However, the RMSE and MAE values were higher than those of the random forest algorithm. Aligned with the results of this study, D'Este et al. [85] evaluated the performance of machine learning models in estimating the fine fuel load in a region of Italy. They observed greater predictive power for the random forest algorithm, with $R^2 = 0.50$, compared to multiple linear regression models and support vector machines, which presented coefficients of determination of 0.40 and 0.39, respectively. Compared with this study, despite different fuel characteristics, the $R^2$ values for estimating the fine dead fuel load (<0.64 cm diameter) using the RF algorithm had a better performance ($R^2 = 0.83$).

The application of the k-NN algorithm to estimate the surface fuel of the plant physiognomy of the Cerrado grassland was unsatisfactory, presenting lower statistical metrics than multiple linear regression analyses and the random forest algorithm. This may be because the k-NN algorithm has a better performance and more comprehensive application in estimating forest variables [69,71,86,87] and due to little knowledge of the assessment of surface fuels.

## 5. Conclusions

Multiple linear regression analyses showed better statistical results for equations adjusted for the beginning of the dry season (May and June) than those adjusted for the end of the dry season (August and September). This behavior arises from the various changes in fuel characteristics in the Cerrado grassland's physiognomy during the dry season's beginning and end. Therefore, modeling to obtain estimates for load, moisture, and other surface fuel characteristics should be performed separately and consider the seasonality throughout the year.

The use of the random forest algorithm contributed to improvements in the evaluation metrics for estimating the Cerrado grassland surface fuel load compared to multiple linear regression analyses and the k-nearest neighbor algorithm. Out of the predictor variables originating from the products of Landsat 8 OLI images, the fraction-soil variable from the spectral mixture analysis exerted the most significant effect on load estimation in the different classes analyzed herein. Accordingly, applying the RF algorithm and the fraction-soil variable is recommended for estimating the Cerrado's fuel load of open or savanna physiognomies. Vegetation indices also played a considerable role in applying the RF algorithm, especially the NDII, GVMI, DER56, NBR, and MSI, which presented higher effect values than the others in the estimates of the analyzed surface fuel variables.

However, their performance in more closed physiognomies with a more significant presence of tree species, for example, is unknown. Further studies must be conducted to verify the feasibility of using the products of different satellite images in different environments, especially in areas more prone to fires in the biome.

**Author Contributions:** Conceptualization, M.M.S. and A.C.B.; methodology, M.M.S., A.C.B. and M.G.; software, E.H.R., A.D.P.D.S. and J.N.C.; validation, M.M.S. and A.D.P.D.S.; formal analysis, D.B., G.R.D.S., A.C.B. and M.G.; investigation, M.M.S., A.D.P.D.S. and J.N.C.; resources, M.G. and A.C.B.; data curation, M.M.S. and A.D.P.D.S.; writing—original draft preparation, M.M.S.; writing—review and editing, M.M.S. and A.C.B.; visualization, E.H.R., G.R.D.S. and D.B.; supervision, A.C.B. and M.G.; project administration, M.M.S. and M.G.; funding acquisition, M.G. and A.C.B. All authors have read and agreed to the published version of the manuscript.

**Funding:** This research was funded by the National Council for Scientific and Technological Development (CNPQ)–PELD Jalapão (grant number 443100/2020-9).

**Data Availability Statement:** Data are available on request from the authors.

**Acknowledgments:** The authors acknowledge the National Council for Scientific and Technological Development (CNPQ) for financial support granted to carry out this research; the Federal University of Tocantins and the Graduate Program in Forestry and Environmental Sciences (PGCFA) for financial support granted for translating this article through public notice N° 012/2021; and the Serra Geral do Tocantins Ecological Station managers, Ana Carolina Sena Barradas, Marco Assis Borges, and Máximo Menezes Costa, for their support in the planning and logistics of field collections.

**Conflicts of Interest:** The authors declare no conflict of interest.

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
