# Peer review of "Estimating the Surface Fuel Load of the Plant Physiognomy of the Cerrado Grassland Using Landsat 8 OLI Products"

_remotesensing, doi:10.3390/rs15235481_

Round 1
Reviewer 1 Report (Previous Reviewer 1)
Comments and Suggestions for Authors
1. Fig1 - Add grid to A,B,C and D also
2. Major disadvantage of the study is novelty in methodology. Author has to justify in detail about novelty in methodology
3.Data used is very old (2017), suggest to work with recent data and present the dynamics
4.Funding - grant number is missing
5. References - All are very old , need to include recent works in this area
Author Response
Please see the attachment.

Reviewer 2 Report (New Reviewer)
Comments and Suggestions for Authors
Dear Authors,
This scientific article represents an important contribution to the science of mapping and monitoring wildfires in grassland environments of the Cerrado biome in Brazil, which can also be applied in other similar ecosystems on Earth.
The methods were detailed, allowing the replication of this study.
Some corrections are necessary.
Specific comments:
Title:
I suggest changing the title:
“Estimating the Surface Fuel Load of grassland environments of the Cerrado biome Using Landsat 8 OLI Products”
Abstract:
The conclusion in the "Abstract" topic does not respond to the 3 main objectives of this article. I suggest reformulating the conclusion.
Introduction:
Line 52: Change "(i) direct measurements of live fuel moisture" to "indirect measurements of live fuel moisture" because they were obtained by remote sensing and not directly from the field, correct? Or specify if they were both.
Materials and Methods:
Line 83: Add " ... Jalapão region, in Brazil".
Line 97: In the Legend or text of figure 1, add information about the letters A, B, C and D, what they represent for the study.
Are they plant phytophysiognomies from the Cerrado biome?
Line 108: I believe the sub-samples area information is wrong.
I believe you meant to say 0.5 m x 0.5 m = 0.25 m², correct?
Line 166: Change the Roman numeral "(xiv) Derivative - band 3 (vd), band 4 (vm) - DER34 [44];" for "(xix)"
Lines 180-181: Specify which algorithm and software was used to perform the atmospheric correction process of images.
Lines 184-185: Specify which software was used to perform the spectral mixture analyses (NPV, GV and soil).
Lines 211-212: Which software was used to perform digital image processing regarding vegetation indices?
Results:
Explain better the letters A and B in Figure 7, in the text (lines 411-416) and in the caption of figure 7 (lines 418-420).
Conclusions:
It is necessary to include a brief conclusion on the performance of the vegetation indices and fraction values (F-values) of the spectral mixture analysis (SMA) in the estimation of the Cerrado grassland surface fuel load.
Author Response
Please see the attachment.

Reviewer 3 Report (New Reviewer)
Comments and Suggestions for Authors
In this paper, multiple linear regression, random forest and nearest neighbor methods are used to estimate the fine surface fuel load using Landsat 8 OLI image data. Major revisions are required before further processing.
1. In line 37, why separate the Rothermel model, as we all know, the Rothermel model is not the only fire spread model that requires fuel load.
2. In the introduction, the importance of fuel load estimation and the previous methods are not comprehensive and specific.
3. In this paper, 30 m resolution landsat data is selected for estimation, why not choose higher resolution data?
4. In this paper, it seems inappropriate to use the random forest method to evaluate the results of importance, and then draw a conclusion. It is suggested that we can choose another method to test, or use the best model effect to test the importance of variables.
Round 2
Reviewer 1 Report (Previous Reviewer 1)
Comments and Suggestions for Authors
1. Fig1 - Add grid to A,B,C and D also
2. the financial support granted for translating this article through public notice N° 012/2021. ---- check this number
Author Response
1. The Grid was added to the distribution quadrants of the data sampling units, as requested by reviewer 1. The figure was updated in the article's Word file and will be updated in the system during the upload process of this round.
2. The number of the financial support notice for the translation of this article was checked and is correct.
Thank you very much for your considerations.
Reviewer 3 Report (New Reviewer)
Comments and Suggestions for Authors
The author explained the questions raised, but did not make any substantive changes, only made some explanations, but I have no additional questions, the final acceptance is up to the editor to decide.
Author Response
We are very happy that you agree with our modifications and explanations made.
Thank you very much for your kind considerations.
This manuscript is a resubmission of an earlier submission. The following is a list of the peer review reports and author responses from that submission.
Round 1
Reviewer 1 Report
Comments and Suggestions for Authors
Comments to authors:
Ø Novelty of the study is to be addressed properly in the abstract and in the introduction section.
Ø Line 104: fuels classification reference to be included.
Ø Line 117: Mention clearly the date of Landsat images and dates of samples.
Ø Line 132/153: Mention tools used for indices and pre-processing.
Ø Figure 4: Split in to two separate figures ( 4a & 4b or 4 & 5 ) indicating legends, scale, direction etc to both figures. Otherwise, you may go as Figure 3 model.
Reviewer 2 Report
Comments and Suggestions for Authors
The spatial distribution of fine combustible loads on the surface is of great significance for fire management and rescue. This study used Landsat 8 OLI images, based on 68 ground-measured sampling points, and used multiple linear regression, random forest, and K-nearest neighbor methods to estimate the fuel load, among which random forest was the best. The research object of this study is grassland, thus it is reasonable to use optical remote sensing images to estimate the fuel load. However, there are still some statements in this study that are not clear and confuse readers. The specific suggestions are as follows.
Introduction:
1. L 32: The definition of fine fuels should be given when it first appeared.
2. L 43-45: This description is inappropriate. In fact, Jonas Franke et al. estimated the fuel load of grassland in the Cerrado in 2018. Also, the authors should do more literature reviews.
Materials and Methods:
3. Please add the sampling date, the weather conditions at the time of sampling, and the spatial distribution of the 68 sampling points.
4. It would be better to draw a sampling diagram, or provide some photos of the sampling site.
5. L 101-114: It is recommended that the sampling categories be presented in table.
6. L 116: Please list the specific time period when the institute obtained remote sensing images.
7. L 132: A total of 20 vegetation indices have been calculated. I’m not sure if all these VIs are suitable for fuel load estimation. Please explain further why these vegetation indices were chosen and why exactly the 20 vegetation indices were chosen.
8. L 153-168: This part is highly similar to what (Franke et al. 2018) did (including the study area), are there any improvements?
9. L 200-203: Please explain how to adjust the equation for the dry season period
10. L 219-222: How did you determine the parameters (ntree, mtry) in a random forest model?
Results:
11. L 261: What are the independent variables used in linear regression? Are the independent variables used by the three methods consistent in this paper?
12. L 263-267: This sentence is too long to read, please rewrite.
13. L 284-286: Why doesn't the random forest model take into account different dry season periods?
14. Figure 3: Please supplement the 1-h spatial distribution map of woody fuel loads. It is more convenient for readers to mark the accuracy index in the figure. This figure is best placed separately after the explanation of the results of the three models.
Discussion:
15. Figure 4: Are there missing data in the northeast of the study area? However, the load distribution of the missing part is shown in Figure 3, which makes the results unreliable.
16. Why choose Landsat 8 OLI instead of Sentinel-2 data?